# Human Umbilical Cord-Mesenchymal Stem Cells Promote Extracellular Matrix Remodeling in Microglia

**DOI:** 10.3390/cells13191665

**Published:** 2024-10-09

**Authors:** Marta Tiffany Lombardo, Martina Gabrielli, Florence Julien-Marsollier, Valérie Faivre, Tifenn Le Charpentier, Cindy Bokobza, Deborah D’Aliberti, Nicola Pelizzi, Camilla Halimi, Silvia Spinelli, Juliette Van Steenwinckel, Elisabetta A. M. Verderio, Pierre Gressens, Rocco Piazza, Claudia Verderio

**Affiliations:** 1Institute of Neuroscience, National Research Council of Italy, Via Raoul Follereau 3, 20854 Vedano al Lambro, Italy; m.lombardo17@campus.unimib.it (M.T.L.); martina.gabrielli@in.cnr.it (M.G.); camilla.halimi@studenti.unimi.it (C.H.); 2School of Medicine and Surgery, University of Milano-Bicocca, Piazza dell’ Ateneo Nuovo 1, 20126 Milan, Italy; 3Inserm, NeuroDiderot, Université Paris Cité, 75019 Paris, France; florence.julien@inserm.fr (F.J.-M.); valerie.faivre@inserm.fr (V.F.); tifenn.lecharpentier@inserm.fr (T.L.C.); cindy.bokobza@inserm.fr (C.B.); juliette.van-steenwinckel@inserm.fr (J.V.S.); pierre.gressens@inserm.fr (P.G.); 4Department of Medicine and Surgery, University of Milan-Bicocca, 20900 Monza, Italy; deborah.daliberti@unimib.it (D.D.); silvia.spinelli@unimib.it (S.S.); rocco.piazza@unimib.it (R.P.); 5CARE Franchise, Chiesi Farmaceutici S.p.A., 43122 Parma, Italy; n.pelizzi@chiesi.com; 6School of Science and Technology, Nottingham Trent University, Nottingham NG11 8NS, UK; elisabetta.verderio-edwards@ntu.ac.uk; 7Department of Biological Sciences (BIGEA), University of Bologna, Via Francesco Selmi 3, 40126 Bologna, Italy

**Keywords:** human umbilical cord mesenchymal stem cells, microglia, neuroinflammation, extracellular matrix, migration

## Abstract

Human mesenchymal stem cells modulate the immune response and are good candidates for cell therapy in neuroinflammatory brain disorders affecting both adult and premature infants. Recent evidence indicates that through their secretome, mesenchymal stem cells direct microglia, brain-resident immune cells, toward pro-regenerative functions, but the mechanisms underlying microglial phenotypic transition are still under investigation. Using an in vitro coculture approach combined with transcriptomic analysis, we identified the extracellular matrix as the most relevant pathway altered by the human mesenchymal stem cell secretome in the response of microglia to inflammatory cytokines. We confirmed extracellular matrix remodeling in microglia exposed to the mesenchymal stem cell secretome via immunofluorescence analysis of the matrix component fibronectin and the extracellular crosslinking enzyme transglutaminase-2. Furthermore, an analysis of hallmark microglial functions revealed that changes in the extracellular matrix enhance ruffle formation by microglia and cell motility. These findings point to extracellular matrix changes, associated plasma membrane remodeling, and enhanced microglial migration as novel mechanisms by which mesenchymal stem cells contribute to the pro-regenerative microglial transition.

## 1. Introduction

Microglia, the resident immune cells of the CNS, are key players in brain development, homeostasis, and diseases. Throughout their life, microglia undergo constant self-renewal, maintaining a steady but heterogeneous population of cells that shift among an array of distinct morphologies [1]. Ramified microglia (“homeostatic”) constantly survey the environment, engulf synapses, and interact with other brain cells to regulate myelin growth/integrity and support neurons [1,2,3]. In response to danger stimuli, such as ATP released from the damage site, microglia adopt a less ramified shape, activate the inflammasome machinery, and release proinflammatory cytokines [4] to adapt their function against insult and maintain homeostasis in brain tissue [5]. 

Microglia are highly plastic and acquire a variety of morphological states and transcriptional phenotypes in a context-dependent manner. Under pathological conditions, including perinatal brain injury, traumatic brain injury (TBI), or local demyelination, microglia acquire many diverse transcriptional states [6,7,8,9]. However, microglia may share a common core signature in response to damage, the disease-associated microglia (DAM) signature [10], characterized by the upregulation of AD genetic risk factors and decreased homeostatic gene expression, which was originally associated with Alzheimer’s disease (AD) [11]. DAM and related transcriptional states [12,13,14,15] clear protein aggregates or dying cells, contributing to neuroinflammation that promotes tissue repair. However, persistent microglial activation can become detrimental during disease progression owing to the loss of essential homeostatic microglial functions. Restoring these homeostatic microglial functions prevents neurodegeneration in an AD mouse model [16], and forced microglial turnover ameliorates deficits induced by TBI [7]. Hence, identifying new tools to counteract microglial overactivation has become a therapeutic goal in neurological disorders [17].

Among the strategies that can drive microglia toward a beneficial phenotype, mesenchymal stem cells (MSCs) are currently being investigated as attractive cell therapies for neuroinflammatory diseases affecting adults as well as neonates [18]. MSCs modulate several immune cell effector functions [19,20,21,22,23] and attenuate the production of reactive oxygen species, promoting the restoration of brain tissue function in preclinical models of neurological disorders, including models of perinatal brain injury [24,25,26,27,28], TBI [20], and neurodegenerative diseases [29,30,31].

The positive effects of MSCs largely rely on their secretome, a mixture of soluble factors and molecules contained within extracellular vesicles (EVs) [32], which create a pro-regenerative brain tissue microenvironment [33]. Through their secretome, MSCs can redirect microglia from proinflammatory to pro-regenerative functions [18]. Specifically, the secretome of MSCs induces the expression of pro-regenerative microglial markers (Ym1 and Arg1 mRNAs) both in vitro [19] and in vivo in a TBI mouse model, promoting sustained recovery [20]. Moreover, when cocultured with MSCs, microglia that respond to inflammatory cytokines release EVs that, when injected into a demyelinated mouse model, promote myelin repair at the sites of myelin lesions [31]. Nevertheless, the molecular mechanisms by which MSCs facilitate the transition of microglia toward homeostatic/pro-regenerative functions remain to be fully elucidated.

In this study, by using a bulk transcriptomic approach, we revealed the activated microglial signaling pathways induced by coculture with human umbilical cord MSCs (h-MSCs), revealing, for the first time, the remodeling of the extracellular matrix (ECM). Additionally, we investigated the impact of h-MSCs on the fundamental functions of microglia in response to cytokines in vitro, including proliferation, phagocytosis, motility, and antigen presentation.

## 2. Materials and Methods

### 2.1. Study Design

The sample size was estimated on the basis of similar studies previously carried out in the laboratory for phenotypic marker expression, phagocytosis, proliferation [31,32], cell motility [33], and transcriptomic analysis. Researchers performing the final analysis were blinded to the treatment groups. No exclusion criteria were predetermined. Samples/data points were excluded from the analysis only if they were identified as outliers via a ROUT test (Q = 1%). The N number (=) specifies the number of independent experiments or analyzed cells.

### 2.2. Human Umbilical Cord Mesenchymal Stem Cells

The frozen umbilical-cord mesenchymal stem cells (h-MSCs) were provided by Chiesi Farmaceutici S.p.A., Parma, Italy, at passage 4. The h-MSCs were cultured according to a standardized protocol provided by the company, consisting of thawing the cells at 37 °C, centrifuging at 500× *g* for 10 min at room temperature (RT), resuspending the cells in h-MSCs CHE2 medium (provided by Chiesi Farmaceutici S.p.A., Parma, Italy) containing 2000 U/L heparin and 5% human platelet lysate (provided by Chiesi Farmaceutici S.p.A., Parma, Italy) followed by plating in flasks at a density of 15,000 viable cells per cm^2^ (1.125 million/t75 flask), and maintaining at 37 °C, 5% CO_2_, 95% humidity for 48 h. Before usage, the cells were washed with 1× PBS, harvested with 3 U/mL trypsin/EDTA (Gibco^®^Thermo Fisher Scientific, Waltham, MA, USA), centrifuged at 300× *g*, and counted with trypan blue. H-MSCs were plated on transwells (0.4 μm pore size filter; Constar, Corning, NY, USA) embedded in TrueGel3D hydrogel (TRUE1, Sigma-Aldrich, St. Louis, MO, USA) prior to coculture with microglia.

### 2.3. Animals

Wild-type C57BL/6 mice and OF1 mice (Charles River Laboratories, Lecco, Italy and L’Arbresle, France) were housed under standardized conditions in the animal house of the University Milano Bicocca and NeuroDiderot, respectively, at 22 °C under a 12 h light–dark cycle with access to food and water ad libitum.

### 2.4. Primary Microglia Culture and Treatment

Mixed glial cultures containing both astrocytes and microglia were established from postnatal day (P)2 C57BL/6 mouse brains of either sex, as previously described [34]. The brains were harvested from one litter, and the hippocampi and cortices were pooled together, mechanically dissociated, and plated in poly-L-lysine-coated (Merck, Darmstadt, Germany) T175 cm^2^ flasks (3 brains/flask). The cells were maintained in a minimal essential medium (MEM, Invitrogen, Thermo Fisher Scientific, Waltham, MA, USA) supplemented with 20% fetal bovine serum (FBS) (Gibco, Thermo Fisher Scientific, Waltham, MA, USA), which optimizes microglial expansion, 5.5 g/L glucose (Merck, Darmstadt, Germany), and 1% antibiotics (penicillin–streptomycin, Gibco, Thermo Fisher Scientific, Waltham, MA, USA). When the astrocytes reached confluency, a medium conditioned from the murine GM-CSF-transfected X63 cells [35] was added at a ratio of 1:20 as a source of GM-CSF to stimulate microglial proliferation, and the microglia were shaken from the mixed cultures by orbital shaking. Forty-eight h after shaking, the microglia were maintained under control conditions (control), treated with inflammatory cytokines alone for 48 h (activated), or immediately cultivated with h-MSCs in transwells (activated + h-MSCs) for 48 h at a microglia-to-h-MSCs ratio of 1:1. Cytokines, i.e., interleukin (IL)-1β (40 ng/mL, Peprotech, Rocky Hill, NJ, USA), interferon (IFN)-γ (50 ng/mL, Peprotech, Rocky Hill, NJ, USA), and TNFα (40 ng/mL, Peprotech, Rocky Hill, NJ, USA), were dissolved in water and prediluted in the medium before they were added to the cells to reach the final concentrations.

Ex vivo microglial cultures were established as previously described [36,37]. Briefly, P8 OF1 naïve animals (males and females) were decapitated, and brains without cerebellum and olfactory bulbs were dissociated via the Neural Tissue Dissociation Kit containing papain and the gentleMACS Octo Dissociator with Heaters (Miltenyi Biotec, Bergisch Gladbach, Germany). Magnetic beads coupled with mouse anti-CD11B antibodies (microglia) were used for cell isolation according to the manufacturer’s protocol (Miltenyi Biotec, Bergisch Gladbach, Germany). Following cell sorting, the CD11B+ cells were suspended in Macrophage Serum free Media (SFM, Gibco, Thermo Fisher Scientific, Waltham, MA, USA) supplemented with 1% penicillin/streptomycin (Gibco, Thermo Fisher Scientific, Waltham, MA, USA) at a concentration of 6.105 cells/mL (0 days in vitro (DIV)) and plated in 24-well plates (0.5 mL/well). The medium was changed at 1 DIV. At 2 DIV, microglia were stimulated with IL-1β (50 ng/mL, Miltenyi Biotec, Bergisch Gladbach, Germany) and IFN-γ (20 ng/mL, Miltenyi Biotec, Bergisch Gladbach, Germany; PBS for the control group). After 3 h of stimulation, h-MSCs-transwells were added on top of the microglia cultures. After 3 or 21 additional hours, the microglial media were removed, and the plates were kept at −80 °C prior to mRNA extraction.

### 2.5. qRT–PCR

Total RNA was isolated from microglia via Direct-zol™ RNA MiniPrep (Zymo Research, Irvine, CA, USA) following the manufacturer’s protocol. cDNA synthesis was performed via a high-capacity cDNA reverse transcription kit (Applied Biosystems, Foster City, CA, USA) with random hexamers as primers. The resulting cDNAs were amplified via the TaqMan^®^ Gene Expression Assay (Applied Biosystems, Foster City, CA, USA) via the QuantStudio™7 (Thermo Fisher Scientific, Waltham, MA, USA) real-time PCR system. The mRNA expression was normalized to the level of Rpl13 (Ribosomal Protein L13) mRNA. The data obtained were quantified via the 2^−ΔΔCT^ method [38]. The list of primers used can be found in Table 1. For ex vivo microglial culture, mRNAs from 24-well plates were extracted via NucleoSpin RNA XS Plus (Macherey-Nagel GmbH & Co., Düren, Germany) according to the manufacturer’s instructions and diluted in 16 μL of RNase-free water. mRNAs were subjected to reverse transcription via a miScript II RT kit (Qiagen^®^, Hilden, Germany) and an iScriptTM cDNA synthesis kit (Bio-Rad^®^, Hercules, CA, USA). qPCR was performed on selected genes (Table 2), which were analyzed with Rpl13a mRNA as the reference gene as previously described [36,37].

### 2.6. RNA Library Preparation

The RNA libraries were prepared via the SMARTer Stranded Total RNA-Seq Kit v2—Pico Input Mammalian Kit (Takara Bio, San Jose, CA, USA). First, the integrity of the RNA samples was assessed via a TapeStation (Agilent Technologies, Santa Clara, CA, USA): all the RNA samples were high-quality, with an RNA integrity number (RIN) between 8 and 10. Subsequently, 10 ng of each high-quality total RNA sample was fragmented and converted to cDNA through a reverse transcription reaction. Barcoded adapters for Illumina sequencing were added through polymerase chain reaction (PCR), and then the PCR products were purified via AMPure XP beads (Beckman Coulter, Pasadena, CA, USA). Library fragments originating from ribosomal RNA (rRNA) and mitochondrial RNA (mtRNA) were depleted via probes specific to mammalian rRNA. Finally, the remaining cDNA fragments were further enriched in a second round of PCR using primers universal to all the libraries, and the amplification products obtained were purified once more to yield the final cDNA library. Library profiles and concentrations were assessed in running samples at a TapeStation (Agilent Technologies, Santa Clara, CA, USA). The final libraries were also quantified via a Qubit fluorometer (Thermo Fisher Scientific, Waltham, MA, USA).

### 2.7. RNA-Seq Data Analysis

RNA-Seq fastq data were generated via an Illumina NovaSeq 6000. Raw fastq sequences were quality-tested via FastQC (https://www.bioinformatics.babraham.ac.uk/projects/fastqc/ accessed on 16 May 2023) and aligned against the GRCm38/mm10 murine assembly via the STAR splice-aware aligner [39] and the quantMode GeneCounts parameter. The indexing and sorting of the Bam alignment files were carried out via SAMtools v. 1.13 [40]. The sorted, indexed bam files were manually inspected via the Integrative Genomics Viewer [41]. Differential gene expression was carried out via DESeq2 v. 1.30 [42]. The Benjamini–Hochberg corrected *p* value was <0.1, which was the threshold used to identify genes as differentially expressed. GSEA was performed with GSEA software v. 4.2.1 (https://www.gsea-msigdb.org/gsea/downloads.jsp accessed on 16 May 2023). Gene sets were considered significantly enriched if the Benjamini–Hochberg corrected *p* value was <0.25 and were prioritized according to their normalized enrichment score (NES).

### 2.8. Immunocytochemical Staining

Microglia cultured on coverslips and exposed or not to h-MSCs were subjected to live staining for 5 min at 37 °C with the microglial marker Isolectin IB4 (IB4 1:100, Alexa 488- or 568-conjugated, Cat #I21411 and #I21412, Thermo Fisher Scientific, Waltham, MA, USA) to delineate the cell surface and IA12 Abs (mouse monoclonal 1:1000) to reveal extracellular TG2 [43]. The cells were then washed, fixed in 4% PFA, and stained with DAPI (1:50,000, Invitrogen, Cat# D1306) to reveal the nuclei, and a 555 Alexa Fluor mouse secondary antibody (1:200, 1 h at RT; Thermo Fisher Scientific, Waltham, MA, USA) was used to detect TG2. Microglia are the main phagocytic cells in the CNS [44] and efficiently internalize antibodies upon live staining at 37 °C. Therefore, long (1 h) exposure to IA12 resulted in both extracellular and intracellular staining for TG2. The cells were then washed, fixed in 4% PFA, and stained with DAPI (1:50,000, Invitrogen, Cat# D1306) to reveal the nuclei, and a 555 Alexa Fluor mouse secondary antibody (1:200, 1 h at RT; Thermo Fisher Scientific, Waltham, MA, USA) was used to detect TG2. For fibronectin (Fn) labeling, microglia were fixed and stained under nonpermeabilizing conditions (without Triton X-100) with anti-Fn (#F3648, Sigma-Aldrich, St. Louis, MO, USA) antibodies overnight at 4 °C and then with 555 Alexa anti-rabbit secondary Abs. MHC-II and Clec7A staining was performed on fixed microglia in permeabilizing conditions, with anti-MHC class II (MHCII, 1:100, #107621, Biolegend, San Diego, CA, USA), Clec7A (cod. mabg-mdect-2, InvivoGen, San Diego, CA, USA), and Alexa Fluor 647 or 555-conjugated anti-rat secondary antibodies. Twelve-fifteen images/coverslips were captured via a 40 × (TG2, Fn, MHC-II, Clec7a) objective on a Zeiss LSM 800 confocal microscope (Zeiss, Oberkochen, Germany). To quantify cellular TG2 and Fn expression, the spaces covered by the cells were set as regions of interest (ROIs), and the total fluorescence of the ROIs was calculated (IntDen) and divided by the number of cells. To quantify TG2 and Fn extracellular expression, the whole cell-empty space in each image was set as the ROI, and the mean fluorescence was measured. A color balance adjustment, which was the same for all the images in an experiment, was applied. To quantify cellular Clec7a and MHC-II expression, corrected total cell fluorescence (CTCF) per field was measured.

### 2.9. Cell Motility and Ruffle Formation

Microglia cultured on coverslips and exposed or not to h-MSCs were live-imaged with a 40× objective using an Axiovert 200 M (Zeiss, Oberkochen, Germany) microscope equipped with a spinning disk system (UltraVIEW, Perkin Elmer, Waltham, MA, USA). Images were acquired every 30 s (2 frames per minute) via the software Volocity 6.3.0 (Perkin Elmer, Waltham, MA, USA). The ImageJ plug-in “MTrackJ” (ImageJ version 1.54f) was used to manually track the migration paths of microglia in time-lapse videos as described previously [33]. To this end, the cell nucleus was identified at each time point in the optical section. For each cell and time point, the software calculated the migration length (mm) and speed (mm/min). Only cells with a cell body within the recording field were considered. For each recorded cell, the maximum number of ruffles present simultaneously was also measured to evaluate ruffle formation.

### 2.10. Proliferation Assay

Microglial proliferation was determined with a Click-iT^®^ EdU Cell Proliferation Kit (Cat# C10338, Thermo Fisher Scientific, Waltham, MA, USA) following the manufacturer’s instructions. Briefly, microglia were cultured with EdU (10 mM) for 24 h and fixed in 4% paraformaldehyde solution (PFA) for 15 min. After being washed, the cells were permeabilized with 0.5% Triton X-100 for 20 min and incubated in Click-iT reaction cocktail. Fifteen fields per coverslip were captured via a 40× objective on a Zeiss LSM 800 confocal microscope (Zeiss, Oberkochen, Germany), and the percentage of EdU-positive/total IB4+ cells was quantified.

### 2.11. Phagocytosis Assay

Fluorescent latex beads (Cat# L2778, Merck, Darmstadt, Germany) were pre-opsonized in FBS (1:5) for 1 h at 37 °C with frequent agitation. The beads containing FBS were diluted with MEM to a final concentration of 0.01% (*v*/*v*) for beads and 0.05% (*v*/*v*) for FBS and incubated with microglia. After 1 h, the microglia were washed with ice-cold phosphate-buffered saline (PBS) 5 times, fixed in 4% PFA, and immuno-stained with DAPI and IB4. Fifteen images/coverslips were captured via a 40× objective on a Zeiss LSM 800 (Zeiss, Oberkochen, Germany) confocal microscope, and the percentage of microglia that engulfed beads over total microglia was determined. The results are expressed as the percentage of phagocytic cells.

## 3. Results

### 3.1. Expression of Immunomodulatory and Inflammatory Markers in Microglia Cocultured with h-MSCs

Mouse microglia isolated from postnatal mixed glial cultures were exposed to inflammatory cytokines (TNFα+IFNγ+IL-1β) and cocultured with or without h-MSCs in transwells. After 48 h, the h-MSCs were removed, and the microglia were processed for qPCR analysis. Untreated microglia cultured alone were used as controls.

To characterize the impact of h-MSCs on activated microglia (TNFα+IFNγ+IL-1β stimulation), we first quantified the expression of protective/immunomodulatory genes (Arg1 and Socs3 mRNAs), inflammatory genes (Tnfα, Ptgs2, Nos2, Il1β, and Il6 mRNAs), the homeostatic gene Tmem119, and the DAM/activated response microglia (ARM) gene Clec7a mRNA via qPCR. As expected, under inflammatory conditions, microglia cultured alone upregulated inflammatory genes (Tnfα, Ptgs2, Nos2, Il6 mRNAs), the immunoregulatory gene Socs3, and the activation marker Clec7a, whereas microglia cocultured with h-MSCs upregulated Arg1 and further upregulated immunoregulatory (Socs3 mRNA) and homeostatic (Tmem119 mRNA) genes and downregulated Clec7a mRNA. The impact of h-MSCs on microglial inflammatory gene expression was complex; some inflammatory genes were downregulated (Tnfα and Ptgs2 mRNAs), whereas others (Nos2, Il6 and Il1β mRNAs) were overexpressed (Figure 1A).

Taken together, these findings show that h-MSCs induce protective marker expression and partially counteract inflammatory gene expression in microglia in response to cytokines, although the cells maintain and even upregulate some inflammatory traits.

To corroborate these findings, we quantified the expression of immunomodulatory markers (Socs3, Il-4ra, and Il-1rn), inflammatory markers (Nos2, Ptgs2, and Tnfα), and the anti-inflammatory marker Igf1 in ex vivo microglial cultures [36]. Ex vivo microglia were stimulated with IL-1β + IFNγ for 3 h before being cocultured with h-MSCs for either 3 or 21 h. IL-1β + IFNγ stimulation induced the overexpression of proinflammatory markers (Nos2, Ptgs2, and Tnfα mRNAs) after 6 and 24 h and immune regulatory markers (Il-4ra and Socs3 mRNAs) 24 h after stimulation. After 3 h of coculture with h-MSCs, there was no modulation of proinflammatory markers, but significant overexpression of the immunoregulatory marker Socs3 was observed. After 21 h of h-MSCs coculture, there was significant overexpression of immunoregulatory markers (Il-4ra, Il-1rn, and Socs3 mRNAs, even higher than at 3 h of coculture), which correlated with a significant downregulation of proinflammatory markers (Ptgs2 and Tnf mRNAs) and the anti-inflammatory marker Igf1.

Collectively, these data indicate that h-MSCs counteract inflammatory gene expression more quickly and efficiently in ex vivo microglia in response to inflammatory cytokines than do microglia obtained from mixed cultures, which is in agreement with the greater reactivity of microglia remaining longer in vitro in the absence of inhibitory signals present in the brain environment [45].

### 3.2. Bulk Transcriptomic Analysis of Microglia Responding to Cytokines Secreted by h-MSCs

To gain more insight into the transcriptional changes induced by h-MSCs transwell coculture with activated microglia, we next performed bulk RNA-seq.

Differential gene expression analysis between activated microglia cultured in the presence or absence of h-MSCs revealed a total of 1519 significantly dysregulated mRNAs (711 upregulated and 808 downregulated, Benjamini–Hochberg adjusted *p* value < 0.1; Figure 2A). Among the genes most highly upregulated by h-MSCs, we identified the protective/immunomodulatory genes Arg1 and Socs3, along with Il1β and serum amyloid A-3 (Saa3), a gene that stimulates inflammasome activation and IL-1β production (Figure 2A). These findings confirmed the results previously observed through qPCR analysis (Figure 1A). In addition, we detected significant upregulation of genes encoding the ECM components thrombospondin-1 (Thbs1) and versican (Vcan) and the ECM regulators vascular endothelial growth factor A (Vegfa), matrix metalloproteinase 9 (Mmp9), and transglutaminase 2 (Tgm2), which are extracellular cross-linkers that stiffen the ECM and are key enzymes involved in cancer cell survival and epithelial–mesenchymal transition [46] (Figure 2A). Conversely, the homeostatic gene Cx3cr1 and the lipoprotein lipase (Lpl) mRNA were among the top downregulated genes (Figure 2A). This finding is in accordance with the literature data showing a downregulation of these markers upon microglial activation [47].

A pathway overrepresentation analysis of 1519 differentially expressed genes (DEGs) highlighted a critical role for microglia in actively modifying the ECM and influencing ECM-dependent neuronal activities, such as axon guidance (Appendix A). In line with these findings, a gene set enrichment analysis (GSEA) revealed that the EPITHELIAL_MESENCHYMAL_TRANSITION (EMT) hallmark was the most enriched gene set (normalized enrichment score (NES) > 2, padj = *p* < 10 × 10^−16^) (Figure 2B,C), confirming that h-MSCs-treated microglia may be involved in ECM remodeling [48] and deposition. To further investigate this signaling pathway, we performed GSEA using the MATRISOME dataset [49], a collection of 10 gene sets specific for the ECM network. Ten out of the ten gene sets were enriched (FDR < 25%) in h-MSCs-treated microglia compared with activated microglia, revealing that ECM gene expression was prominently altered by h-MSCs (Appendix A). The most enriched gene set was NABA_CORE_MATRISOME, an ensemble of genes encoding core extracellular matrix elements, including ECM glycoproteins, collagens, and proteoglycans [50] (Figure 2D). Similar results were observed when h-MSCs-treated activated microglia were compared to control microglia (10/10 gene sets enriched; Figure 2E and Appendix A) but not when activated microglia were compared to control cells (1/10 gene set enriched, i.e., NABA_SECRETED_FACTORS; Appendix A).

Taken together, these data strongly support the notion that changes in the ECM caused by microglia are directly modulated by h-MSCs treatment.

In-depth analysis of the NABA_CORE_MATRISOME leading edge, composed of those genes supporting the statistical enrichment of the matrisome gene set, revealed the presence of genes encoding several critical ECM components, e.g., collagen isoforms, laminin 1–2–4, vitronectin, fibronectin type III domain containing 8 (Fndc8), and several members of the TGFβ chemokine family (Tgfbi, Tgfb2, Tgfb3), which are known to be potent inducers of ECM proteins and protease inhibitors that prevent ECM degradation (Appendix A). Other upregulated NABA genes included transglutaminases (Tgm1–2–4–6–7), which play important roles in ECM stabilization and resistance to degradation through extracellular cross-linking activity. Collectively, the observed NABA gene upregulation suggested increased ECM deposition and stabilization.

Other top enriched hallmark gene sets identified by GSEA between hMSCs-treated microglia and activated microglia cultured alone were related to the inflammatory response (NES 1.54; FDR: 0.053), TNFα signaling (NES 1.91; FDR: 0.001), and IL6/JAK/STAT3 signaling (NES 1.77; FDR: 0.007) (Figure 2C), indicating increased microglial inflammation. However, the pathway analysis and GSEA also revealed enriched signaling by the immunoregulatory cytokines interleukin-4 and -13 and by interleukin-10, a cytokine that plays a critical role in limiting inflammatory responses in hMSCs-treated microglia compared with activated microglia cultured alone (Figure 3A,B). In addition, further GSEA using gene sets specific for the negative regulation of immune processes confirmed the enrichment of genes involved in immune regulatory pathways (Figure 3B), which is in line with the complex action of h-MSCs on the microglial inflammatory response (Figure 1A).

The top downregulated hallmarks in hMSCs-treated microglia compared with activated microglia were instead associated with impaired replication and cell division (Figure 2B,C). The GSEA of gene sets specific for key microglial functions confirmed the downregulation of genes controlling the cell cycle (Figure 3C), the downregulation of antigen presentation-related genes (Figure 3C and Appendix A), and the lack of changes in the expression of genes involved in phagocytosis. Conversely, genes related to focal adhesion, chemokine activity, myeloid cell migration, and motility were significantly enriched in hMSCs-treated microglia compared with activated microglia cultured alone (Figure 3C), suggesting that h-MSCs treatment improves the capacity of microglia to patrol the environment.

Taken together, these data point to proliferation, antigen presentation, and motility/migration as key microglial functions influenced by h-MSCs, in addition to ECM remodeling and the modulation of the inflammatory response.

### 3.3. h-MSCs Reduce MHCII Expression While Increasing ECM Deposition and Motility in Activated Microglia

To explore how gene expression changes induced by h-MSCs impact microglial function, we next processed activated microglia cultured alone or with h-MSCs for analysis of ECM deposition, cell motility, proliferation, phagocytosis, and antigen-presenting capacity.

Immunofluorescence analysis of the ECM component fibronectin revealed increased staining in h-MSCs-treated microglia and the surrounding environment compared with that in activated microglia. Fibronectin labeling was approximately 7.4- and 5-fold greater at the cell surface and outside the cell, respectively (Figure 4A,B). Immunoreactivity for the extracellular crosslinker TG2 was also greater, both intracellularly (approximately 2.2-fold) and extracellularly (approximately 5.2-fold) (Figure 4C,D). As TG2 is externalized in the extracellular space, which favors its activity (high Ca^2+^/GTP ratio), these data suggest an increase in TG2 enzyme activity and ECM remodeling/stabilization in activated microglia exposed to h-MSCs.

To monitor microglial motility, the cells were subjected to time-lapse imaging every 30 s for 20 min, the distance covered by the cells was measured, and the mean number of actin-rich membrane protrusions (ruffles) per cell was quantified [33]. The analysis revealed greater path length and speed in microglia cultured with h-MSCs than in those cultured alone and increased cell ruffle formation (Figure 4E,F), a characteristic feature of actively migrating cells.

Microglial proliferation was monitored by exposing the cells to the thymidine analog EdU, which becomes incorporated into the DNA of dividing cells. The percentage of EdU + proliferating microglia was strongly decreased in activated microglia compared with control cells, which is consistent with previous studies showing that the inflammatory stimulus LPS strongly inhibited microglial proliferation [51] but was not further decreased by h-MSCs (Figure 5A), excluding the major effect of h-MSCs on this cell function.

Microglial phagocytosis was analyzed by incubating the cells with fluorescent latex beads and quantifying their microglial engulfment via immunofluorescence analysis. The phagocytic capacity of activated microglia cultured alone or cocultured with h-MSCs did not differ from that of control cells (Figure 5B).

Finally, we evaluated major histocompatibility complex II (MHCII) protein expression, an index of microglial antigen-presenting capacity, and the expression of CLEC7a, a marker of activated response microglia (ARMs). As expected, MHCII and CLEC7a immunoreactivity was increased in microglia that responded to inflammatory cytokines compared with control microglia cultured alone (Figure 5C,D), indicating increased antigen-presenting capacity and cell activation. However, MHCII levels returned to control levels in activated microglia cocultured with h-MSCs (Figure 5C), and CLEC7a expression was reduced even below control levels (Figure 5D), revealing the ability of the MSCs secretome to dampen immune microglial function.

Collectively, these findings show that h-MSCs promote ECM deposition and motility in microglia in response to inflammatory stimuli but do not significantly impact the proliferative and phagocytic capacity of these cells. In addition, h-MSCs counteract the increase in MHCII and CLEC7a protein levels induced by inflammatory cytokines.

## 4. Discussion

Efforts have been made in recent years to show that microglia that respond to inflammatory cytokines switch from detrimental to pro-regenerative functions in response to the secretome of MSCs [52,53], but the molecular mechanism(s) underlying the phenotypic microglial transition remain largely undefined.

In this study, we exploited bulk transcriptomic analysis of activated microglia in noncontact cultures with h-MSCs to identify the molecular pathways affected by h-MSCs and explore their impact on microglial functions. We showed that the most relevant pathway altered by the h-MSCs secretome is related to ECM remodeling, leading to increased ECM deposition and increased microglial motility.

Among the top upregulated genes in activated microglia exposed to the h-MSCs secretome, we identified genes encoding thrombospondin-1, an ECM protein with anti-inflammatory properties [54,55]; versican, a structural ECM element and a marker of ECM remodeling [56,57]; and transglutaminase 2 (TG2), an ECM-modifying enzyme that may increase ECM stiffness. GSEA using hallmark gene sets and the MATRISOME gene sets confirmed the upregulation of genes encoding ECM components or modulators, including isoforms of TGFβ, the most well-studied and potent promoter of ECM deposition [58,59]. Importantly, by immunofluorescence analysis, we showed that changes in ECM gene expression result in modifications of the ECM composition in microglia, as evidenced by increased levels of fibronectin, an interstitial matrix component that surrounds cells, and TG2. Several growth factors have been identified in the secretome of MSCs that may directly induce ECM changes in cocultured microglia, including the growth factors HGF, VEGF, βFGF, IGF, and TGFβ [56,57,60]. Interestingly, among these factors, TGFβ induces the expression of both ECM genes and ECM-modifying genes, including collagen and TG2 [61], thus representing a candidate molecule responsible for MSCs-dependent transcriptional ECM changes in microglia. Specifically TGFβ receptor signaling through the Smad3 pathway and Smad3/Smad4 complex translocation to the nucleus regulates collagen gene expression and synthesis of many ECM proteins [62]. Moreover a TGF-β response element in the promoter region of TGM2 (TG2 gene) mediates the regulation of the gene expression specifically [59].

Previous studies have demonstrated that MSCs infusion can induce ECM rearrangements in an experimental model of stroke and amyotrophic lateral sclerosis (ALS) [63,64], leading to improved functional recovery. Specifically, after stroke, MSCs induce a reduction in perineural nets (PNNs) and ECM structures enwrapping the soma of PV-positive interneurons, allowing for restorative plasticity through circuit reorganization in the perilesional cortex [63], whereas in ALS rats, MSCs promote PNN preservation, accounting for better survival of motor neurons [64]. However, whether brain cells such as microglia actively contribute to MSCs-induced ECM remodeling has not yet been investigated [65]. Our study suggests that microglia may act as key players in the ECM remodeling induced by h-MSCs therapy, contributing to beneficial outcomes in neuroinflammatory pathologies. Accordingly, recent evidence has shown that microglia play a role in PNN homeostasis and that remodeling of the brain ECM via phagocytosis or the release of ECM-degrading enzymes is a fundamental microglial function in both physiological and pathological conditions [66,67].

ECM alterations, including PNN disruption, are present in models of perinatal brain injury [68,69] as well as in late-onset neurodegenerative diseases such as Alzheimer’s disease (AD) and Huntington disease (HD) [70,71] and contribute to adverse neurological outcomes of these brain disorders characterized by microglial activation. Perinatal brain damage and AD and HD pathologies benefit from MSCs cell therapy by intravenous transplantation [72,73] or intranasal administration [26,74,75,76], a non-invasive delivery route that allows for MSC penetration into injured brains [77]. Whether ECM alterations can be rescued by intranasal MSCs cell therapy and whether microglia can contribute to MSCs-induced ECM remodeling in these pathologies are relevant questions for future investigations.

Cell motility, a process regulated by the ECM, is essential for the ability of microglia to survey the brain parenchyma and migrate toward lesion sites. Here, we show that this key microglial function is controlled by the h-MSCs secretome. At the molecular level, microglial motility depends on reorganization of the actin cytoskeleton and chemokine signaling. Accordingly, our transcriptomic data revealed that h-MSCs-treated microglia are enriched in pathways involved in focal adhesion formation, chemokine activity, and cell motility and migration.

The top genes contributing to these pathways include VEGFA, which regulates focal adhesion assembly [78], and caveolin 1 and 3, which control anchorage-dependent signaling [79]; C-C or C-X motif chemokine ligands; ECM-associated molecules, including growth factors, chemokines, and growth factor receptors; and molecules involved in adhesion to other cells or to the ECM or in repulsion from adhesion, such as semaphorins and ephrin receptors. Other relevant genes that may promote cellular motility and control lamellipodia formation include RhoG, a small GTPase that regulates focal adhesion formation and turnover [80], several Rho GTPase-activating proteins, ARHGAPs, and the Rho GTPase CDC42 binding protein G (CDC42BPG), which may promote lamellipodia formation downstream of CDC42, as reported elsewhere after microglia are exposed to MSCs [81]. These transcriptional changes, possibly induced by CSF-1, a previously identified factor in the secretome of adipose-tissue-derived MSCs [82], lead to the formation of more ruffles, which may have indirect promigratory functions [81], and to faster cell motion, as indicated by time lapse imaging. Notably, h-MSCs-treated microglia move faster in an ECM that might have become stiffer as a consequence of enhanced transglutaminase crosslinking activities and increased fibronectin deposition. This is in line with the well-known role of matrix stiffness in regulating cell behavior, including cell migration [83], and with recent evidence showing that cells adhere more strongly to substrates and move faster in a thickened environment, a consequence of the change in the mechanical (not biochemical) properties of the microenvironment [84]. Exploring whether changes in microglial motility occur in preclinical models of neuroinflammatory diseases upon MSCs therapy is worth further investigation, as the movement of microglial processes allows microglia to survey the environment, clear cellular debris, remodel the ECM, and interact with neurons and synapses [85].

Finally, our transcriptional data and immunofluorescence analysis revealed that the secretome of h-MSCs has complex effects on the microglial response to inflammatory cytokines. GSEA of hallmark gene sets and inflammation-specific gene sets revealed that inflammatory pathways are further activated in microglia in response to cytokines cocultured with h-MSCs than in those cultured alone. Nevertheless, the protective gene Arg1, which encodes a microglial cluster that is abundant during early brain development and is crucial for the maturation of cholinergic neuronal circuits and cognition [86], is the top upregulated gene in h-MSCs-treated microglia. Furthermore, h-MSCs-treated microglia are enriched in the IL-4, IL-13 and IL-10 anti-inflammatory signaling pathways, depleted of antigen-presenting genes and characterized by decreased MHC-II protein expression. These immunomodulatory effects may be due to MSCs-derived IL-4 and IL-10, whose production is amplified in response to high levels of inflammatory cytokines [59].

The co-upregulation of proinflammatory and anti-inflammatory genes was previously reported by our group in rat microglia exposed in vitro to bone marrow-derived murine MSCs and was associated with the acquisition of pro-regenerative microglial functions, i.e., the ability to promote the differentiation of oligodendrocytes in vitro and at the site of myelin lesions [31,87]. Along with these previous findings, the present transcriptomics data suggest that h-MSCs induce a beneficial microglial phenotype through balancing pro-inflammatory and anti-inflammatory responses. This is in line with the general ability of MSCs to perform their therapeutic function via a balance of pro-inflammatory and anti-inflammatory secretory signals [59,88]. Thus, further activation of specific microglial inflammatory pathways may not be necessarily detrimental and may even contribute to promote brain repair. On the other hand, the significant downregulation of a set of inflammatory genes induced by h-MSCs in ex vivo microglia, which are more closely related to microglial physiology, suggests that in vivo, h-MSCs may have a greater ability to counteract the inflammatory microglial response. Therefore, our results cannot rule out the possibility that the augmented expression of pro-inflammatory genes in microglia exposed to h-MSCs in vitro may be due to the widespread responsiveness of microglia removed from their normal environment [45].

## 5. Conclusions

To conclude, via an in vitro coculture approach combined with transcriptomic and functional analyses, we revealed that ECM remodeling and cell motility are novel functions governed by the h-MSCs secretome in microglia. Transcriptional changes controlling ECM remodeling may be the key to pro-regenerative microglial transition, which contributes to the protective effects of h-MSCs in experimental models of neuroinflammatory diseases, a hypothesis that remains to be explored in future studies.

## Figures and Tables

**Figure 1 cells-13-01665-f001:**
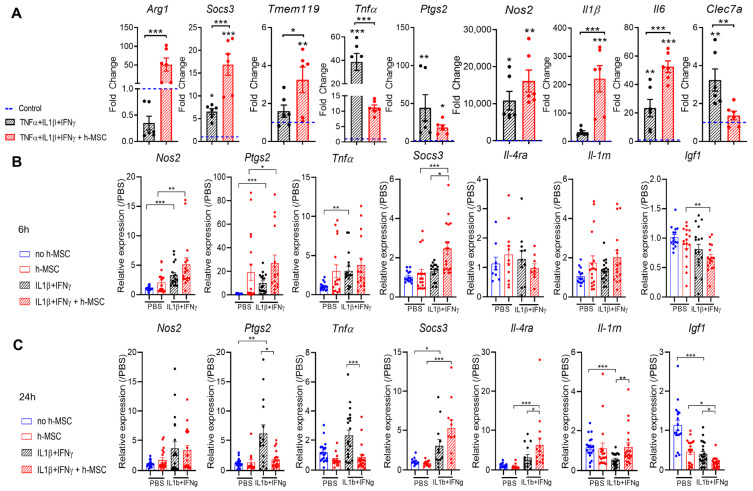
h-MSCs affect homeostatic and activation marker expression in primary mouse microglia. (**A**) qPCR analysis of primary mouse microglia exposed to inflammatory cytokines (IL1β, TNFα, and IFNγ) and cocultured with h-MSCs for 48 h in transwells. Compared with those in activated cells, the expression of the proregenerative marker Arg1, the immunomodulating marker Socs3, and the homeostatic marker Tmem119 was significantly increased after h-MSCs exposure. The data are presented as the means ± SEs normalized to nonstimulated cells (fold change of 1). (Kruskal–Wallis multiple comparisons test: Arg1 *p* = 0.007; one–way ANOVA with Tukey’s multiple comparisons test: Socs3 *p* < 0.001, Tmem119 *p* = 0.04; N = 6). The activation marker Clec7a and the inflammatory marker Tnfα are downregulated (one-way ANOVA, Tukey’s multiple comparisons test: Clec7a, *p* = 0.006; Tnfα, *p* < 0.001, N = 6), whereas the other inflammatory markers Il1β and Il6 are significantly upregulated in h-MSCs-treated microglia compared with activated microglia (one-way ANOVA, Tukey’s multiple comparisons test: Il1β, *p* < 0.001; Il6, *p* < 0.001, N = 6). (**B**) qPCR analysis of primary mouse microglia magnetically isolated from neonatal mice exposed to inflammatory cytokines (IL-1β, IFNγ) for 3 h and then cocultured with h-MSCs for an additional 3 h. (**C**) qPCR analysis of primary mouse microglia magnetically isolated from neonatal mice exposed to inflammatory cytokines (IL-1β, IFNγ) for 3 h and then cocultured with h-MSCs for an additional 21 h. * *p* < 0.05, ** *p* < 0.01, *** *p* < 0.001.

**Figure 2 cells-13-01665-f002:**
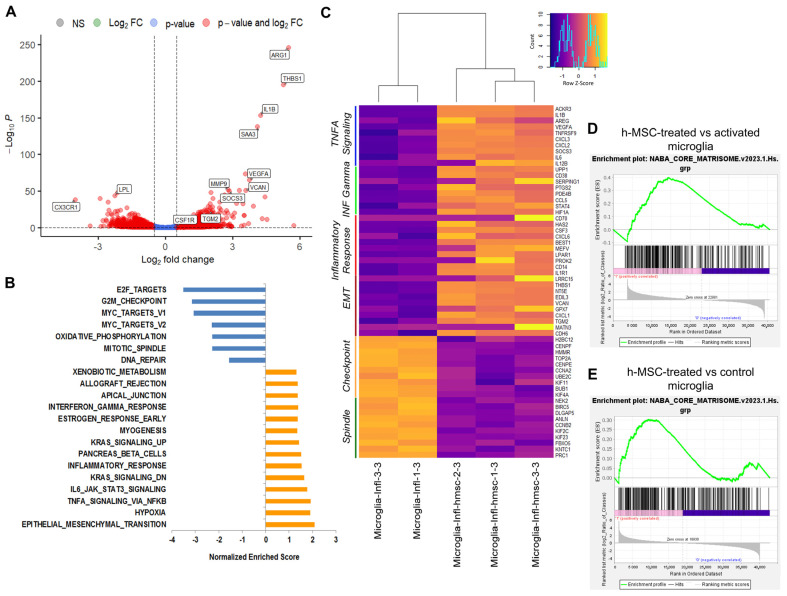
RNA-seq analysis showing extracellular matrix remodeling in microglia cocultured with h-MSCs. (**A**) Volcano plot of differentially expressed genes in h-MSCs-treated activated microglia vs. activated microglia. Red circles represent genes with a |log2-fold change| >  0.5 and a -log10 Benjamini–Hochberg (BH)-adjusted *p* value > 1; blue circles represent genes with a -log10 BH-adjusted *p* value > 1; green circles represent genes with a |log2-fold change| > 0.5; gray circles represent genes with a |log2-fold change| ≤ 0.5 and a -log10 BH-adjusted *p* value ≤ 1. (**B**) Bar plot showing all the significant hallmarks identified via GSEA. Orange bars represent positive enrichment; blue bars represent negative enrichment. (**C**) Heatmap reporting the expression levels of the top leading genes of the 4 positively enriched gene sets and 2 negatively enriched gene sets. TNFα signaling, blue; INF-γ, green; inflammatory response, red; EMT, violet; G2M checkpoint, yellow; mitotic spindle, dark green. (**D**,**E**) GSEA plots reporting the top enriched MATRISOME gene set (NABA_CORE_MATRISOME) in h-MSCs-treated activated microglia vs. activated microglia (**D**) and h-MSCs-treated activated microglia vs. control microglia (**E**).

**Figure 3 cells-13-01665-f003:**
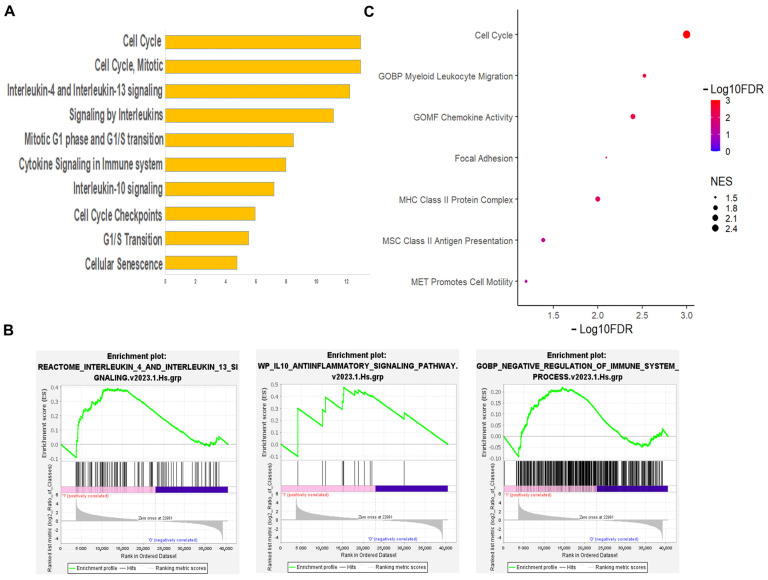
RNA-seq analysis showing cell cycle and immune system regulation by microglia cocultured with h-MSCs. (**A**) Bar plot showing the top 10 Reactome pathways enriched in h-MSCs-treated activated microglia vs. activated microglia. (**B**) GSEA plots reporting gene sets related to anti-inflammatory cytokine signaling pathways (IL4/IL13 and IL10) and negative regulation of immune system processes in activated h-MSCs-treated microglia vs. activated microglia. (**C**) Dot plot of positively or negatively enriched GSEA gene sets related to relevant microglial functions.

**Figure 4 cells-13-01665-f004:**
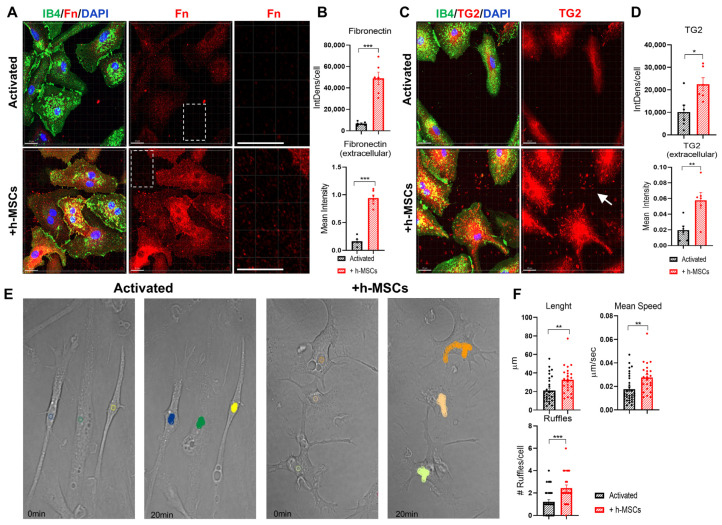
Impact of h-MSCs on extracellular matrix composition and microglial motility. (**A**) Representative confocal z-stack projections by Imaris of activated microglia, cultured with or without h-MSCs, and stained for Fn (red), IB4 (green), and DAPI under nonpermeabilizing conditions. Dotted boxes indicate the zoomed region on the right. Scale bars: 20 mm. Scale bar: zoom-in: 20 mm. (**B**) Corresponding quantification of surface Fn (top; *t*-test, *p* < 0.001, N = 6) and extracellular Fn (bottom; *t*-test, *p* < 0.001, N = 6). (**C**,**D**) Microglia described in A were subjected to live staining with TG2 for 1 h (red) and IB4 for 5 min (green), fixed and stained with DAPI. (**C**) Representative confocal z-stack projections generated by Imaris showing cellular and extracellular (arrow) TG2 staining. (**D**) Corresponding quantification of cellular TG2 (top; *t*-test, *p* = 0.0152, N = 6) and extracellular TG2 (bottom; *t*-test, *p* = 0.0072, N = 6). (**E**) Representative bright fields of microglia shown in A; live images at 0 and 20 min are shown. The colored traces indicate the paths traveled by the cells. The arrows point to membrane ruffles. (**F**) Histograms showing the path length, average speed, and number of ruffles of the imaged cells (Mann–Whitney test, *p* = 0.0026; path length, *p* = 0.0023; average speed, *p* = 0.0008; ruffles, activated N = 32, activated + h-MSCs N = 23). * *p* < 0.05, ** *p* < 0.01, *** *p* < 0.001.

**Figure 5 cells-13-01665-f005:**
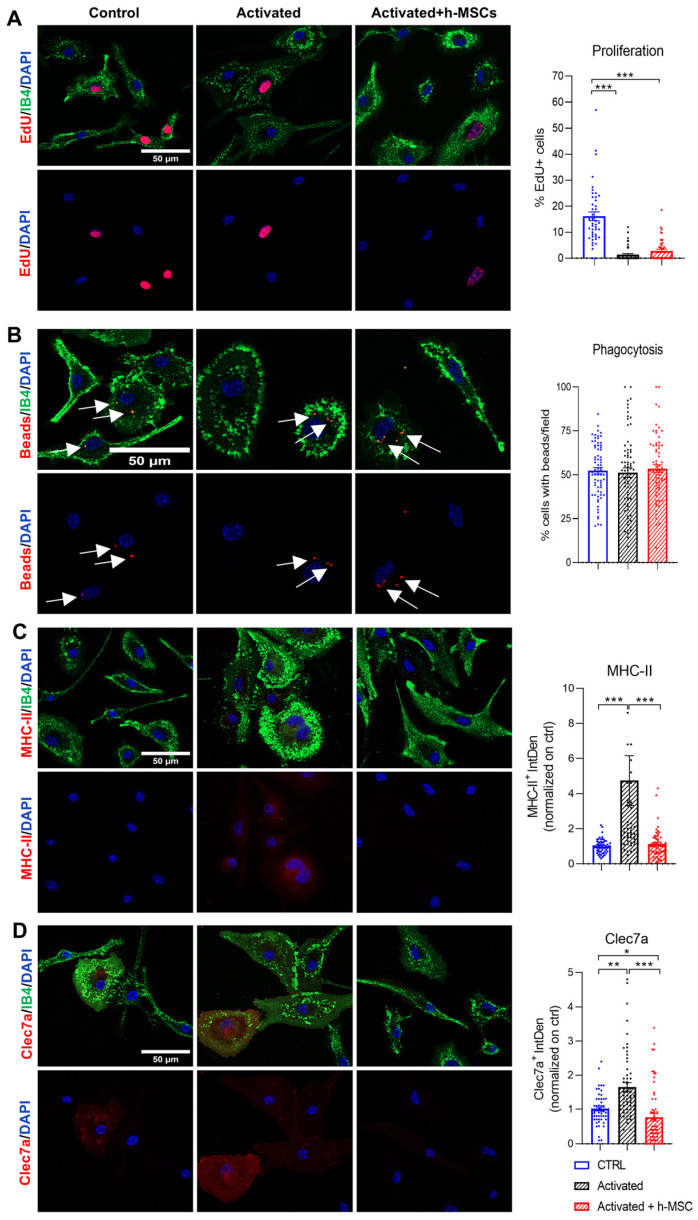
Effects of h-MSCs on microglial proliferation, phagocytosis, and MCHII and Clec7a expression. (**A**) Representative confocal images of control, activated, and h-MSCs-treated activated proliferating microglia that were positive for EdU and double-stained for IB4 (green) and DAPI (blue). On the right, the corresponding quantification is shown. Each dot represents the percentage of proliferating cells in a single field. The data are presented as the means ± SEMs (one-way ANOVA, Tukey’s multiple comparisons test, *p* < 0.001 ctr vs. activated, *p* < 0.001 h-MSCs vs. ctrl, *p* = 0.60 h-MSCs vs. activated, N = 3). (**B**) Representative images of microglia that engulfed fluorescent beads (arrows) under the conditions described in (**A**) and were stained with IB4 (green) and DAPI (blue). On the right, the data are quantified (one-way ANOVA, Tukey’s multiple comparisons test, *p* = 0.59 ctr vs. activated, *p* = 0.20 h-MSCs vs. activated, N = 4). (**C**) Representative confocal images of microglia as in A stained for MHC-II (red), IB4 (green), and DAPI (blue). On the right, CTCF quantification of MHC-II+ microglia per field are normalized to those in the control group (one-way ANOVA, Tukey’s multiple comparisons test, *p* = 0.001 ctr vs. activated, *p* < 0.001 h-MSCs vs. activated, N = 4). (**D**) Representative images of microglia stained with Clec7a (red), IB4 (green), and DAPI (blue) under the conditions described in A and the corresponding CTCF quantification of Clec7a+ microglia per field, normalized to the control (one-way ANOVA, Tukey’s multiple comparisons test *p* = 0.001 ctr vs. activated, *p* < 0.001 h-MSCs vs. activated, N = 4). * *p* < 0.05, ** *p* < 0.01, *** *p* < 0.001.

**Table 1 cells-13-01665-t001:** List of primers used for microglia from mixed cultures.

Gene Symbol	Name	Taqman Assay
Arg 1	Arginase 1	Mm00475988_m1
Ptgs2	Prostaglandin-Endoperoxide Synthase 2/COX-2	Mm00478374_m1
Tmem119	Transmembrane Protein 119	Mm00525305_m1
IL1-β	Interleukin 1-β	Mm00434228
IL6	Interleukin 6	Mm00446190
NOS2	Nitric Oxide Synthase 2	Mm00440502
Rpl13	Ribosomal Protein L13	Mm02526700
Socs-3	Suppressor Of Cytokine Signaling 3	Mm00545913
TNF-α	Tumor Necrosis Factor	Mm00443258
Clec7A	C-type lectin domain containing 7A/dectin-1	Mm01183349

**Table 2 cells-13-01665-t002:** List of primers used for ex vivo microglia.

Gene Symbol	Name	Forward	Reverse
Igf-1	Insulin like growth factor 1	TGG ATG CTC TTC AGT TCG TG	GCA ACA CTC ATC CAC AAT GC
Il1-rn	Interleukin 1 receptor antagonist	TTG TGC CAA GTC TGG AGA TG	TTC TCA GAG CGG ATG AAG GT
Il4-ra	Interleukin 4 receptor antagonist	GGA TAA GCA GAC CCG AAG C	ACT CTG GAG AGA CTT GGT TGG
Nos2	Nitric oxide synthase 2	CCC TTC AAT GGT TGG TAC ATG G	ATC TCC GTG ACA GCC
Ptgs2	Prostaglandin endoperoxide synthase 2	TCA TTC ACC AGA CAG ATT GCT	AAG CGT TTG CGG TAC TCA TT
Rpl13a	Ribosomal protein L13a	ACA GCC ACT CTG GAG GAG AA	GAG TCC GTT GGT CTT GAG GA
Socs3	Suppressor of cytokine 3	CGT TGA CAG TCT TCC GAC AA	TAT TCT GGG GGC GAG AAG AT
Tnf	Tumor necrosis factor	GCC TCT TCT CAT TCC TGC TT	AGG GTC TGG GCC ATA GAA CT

## Data Availability

The data supporting the results of this study are available from the corresponding author on reasonable request. Data are located in controlled access data storage at the repository SRA, ID: PRJNA1157811, https://www.ncbi.nlm.nih.gov/sra/PRJNA1157811.

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
