# Peer review of "Human Umbilical Cord-Mesenchymal Stem Cells Promote Extracellular Matrix Remodeling in Microglia"

_cells, 2024, doi:10.3390/cells13191665_

Round 1

Reviewer 1 Report

Comments and Suggestions for Authors

It is a well-designed and presented study on the influence of mesenchymal cells on microglia. The data indicate that the secretion of extracellular matrix proteins is modulated. This has been shown by fluorescence microscopy of fibronectin and gene expression analysis. Significant upregulation of ECM-related genes such as fibronectin, thrombospondin-1, and transglutaminase-2 (TG2) indicated increased ECM deposition and remodeling. Some controversial data are presented considering the complex inflammatory gene expression since some inflammatory genes were downregulated and others were overexpressed.  This discrepancy should be further explained in the discussion section. The feasibility of in vivo therapeutic use of MSCs should also be discussed. Can MSCs pass the Blood Brain Barrier? How can these cells reach the brain in vivo?

Reviewer 2 Report

Comments and Suggestions for Authors

In this paper, the authors show that substances secreted by umbilical cord-derived mesenchymal stem cells act on microglia and induce complex responses, including the production of inflammatory molecules and anti-inflammatory effects. In particular, a novel finding is that microglia exposed to the secretome of mesenchymal stem cells undergo extracellular matrix remodeling. However, the manuscript still seems to be in need of many revisions for publication.

1. In previous studies, it is known that extracellular vesicles and soluble factors such as HGF of mesenchymal stem cells act on microglia. No specific substances have been shown in this study. The complex responsiveness described earlier may also be due to the fact that we are looking at the responsiveness of multiple substances. Candidate substances that contribute to changes in the extracellular matrix should be indicated.

2. It is not known whether the microglial changes shown in this study are specific in co-culture with mesenchymal stem cells. Comparisons with other cells should provide evidence of specificity.

3. Are there differences in results between mesenchymal stem cells derived from umbilical cord and those from other sources?

4. When I read this paper, it was unclear what therapeutic application the authors had in mind. If the authors are only thinking of cell transplantation, the contrast with previous studies should be mentioned a little more in the introduction and discussion. For example, to what extent can mesenchymal stem cells be delivered to the brain parenchyma by intravenous mesenchymal stem cell transplantation? Or do you envisage transplanting mesenchymal stem cells directly into the brain parenchyma?

Round 2

Reviewer 2 Report

Comments and Suggestions for Authors

This manuscript has been revised well and is in a nice condition now.